



# Measurement Report: Sulfuric Acid Nucleation and Experimental Conditions in a Photolytic Flow Reactor

David R. Hanson, Seakh Menheer, Michael Wentzel, Joan Kunz

Chemistry Department, Augsburg University, Minneapolis, MN

*Correspondence to*: David R. Hanson (hansondr@augsburg.edu)

**Abstract.** Nucleation involving sulfuric acid and water has been studied in a photolytic flow reactor over a time period of several years. Results show that the system - flow reactor, gas supplies and lines, flow meters, valves, $H_2SO_4$ photo-oxidant sources – has a baseline stability that yields nucleation information such as cluster free energies. The baseline nucleation rate is punctuated by temporary bursts that in many instances are linked to cylinder changes, delineating this source of

potential contaminants. Diagnostics were performed to better understand the system include growth studies to assess $H_2SO_4$ levels, chemiluminescent NO and $NO_x$ detection to assess the HONO source, and deployment of a second particle detector to assess the nanoparticle detection system. The growth studies show trends consistent with calculated $H_2SO_4$ levels and also provide an anchor for $H_2SO_4$ concentrations. The chemiluminescent detector revealed that small amounts of NO are present in the HONO source, ~10 % of HONO. The second, condensation-type particle counter indicates that the nanoparticle sizing

system has a bias at low sulfuric acid levels. Modeling studies yield nucleation rates as a function of sulfuric acid concentration that probably represent upper limits to nucleation in the binary system, $H_2SO_4$-$H_2O$, as contaminants might act to enhance nucleation rates or ion-mediated nucleation may contribute. Nonetheless, the experimental nucleation rates are some of the lowest reported so far in experiments where sulfuric acid is photolytically-generated. Results from experiments with varying water content and with ammonia addition are also described. The energetics of clusters in this three component

system reveal a challenging interplay between the components (e.g. previously reported ion-mediated and homogeneous rates are unintentionally similar) and experiments indicate water plays a significant role in nucleation involving sulfuric acid and ammonia.

## 1 Introduction

Atmospheric nucleation involving sulfuric acid, water and ammonia is believed to have a large impact on the properties of

clouds and their effects on the radiation balance of the atmosphere [Dunne et al. 2016; Coffman and Hegg, 1995]. Since the influence on climate of aerosol particles is well-known to be potentially quite large [IPCC, 2013], the formation of atmospheric particles through nucleation has long been studied with a focus on sulfuric acid [McMurry et al., 2005; Kulmala et al. 2004]. The chemical systems thought to have the largest global impacts on new particle formation are the binary $H_2SO_4$-$H_2O$ and ternary $H_2SO_4$-$H_2O$-$NH_3$ systems.

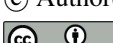



Yet over the decades there is a wide divergence in nucleation rates measured in laboratory experiments, particularly in the binary system - see figure 6 in Zollner et al. [2012] and figure 9 in Hanson et al. [2019]. While many of these discrepancies may be attributable to contamination or other experimental conditions, this leaves nucleation rates for atmospheric conditions somewhat uncertain. Likewise, nucleation rates in the ternary system derived from laboratory experiments have uncertain applicability to the atmosphere due in part to effects of potential contaminant bases.

Recent nucleation results from the CLOUD experiment [Kirkby et al. 2011; Almeida et al. 2013; Ehrhart et al. 2016; Kürten et al. 2016] may best represent binary and ternary nucleation and these results have been parameterized and used in global climate models where their climate effects were found to be significant. Yet these nucleation rates have not been corroborated in independent laboratory investigations. In fact there is disagreement with other recent work, albeit based on extrapolated data and/or theoretical treatments (see Zollner et al. 2012; Hanson et al. 2019; Kürten et al. 2016) .

For the binary system, Ehrhart et al. [2016] showed that the CLOUD data at low temperatures agrees well with theoretical nucleation rates from the SAWNUC thermodynamics [Lovejoy et al. 2004]. However, there was poor agreement at temperatures warmer than 270 K. This poor agreement has been attributed to contamination by base molecules that are enhanced at warm chamber temperatures. Certainly contamination is a concern for experimental work in the binary system [e.g. Zollner et al. 2012; Hanson et al. 2019]. Here, we report more measurements in the putative binary system to gain

further information on the effects of potential contaminants in our Photolytic Flow Reactor (PhoFR) [Hanson et al. 2019]. We present measurements of NO and $NO_x$, growth measurements, and measurements from a second particle counter that improve knowledge of the conditions in PhoFR.

For the ternary system, Kürten [2019] and Hanson et al. [2019] used cluster models to derive thermodynamic information from the CLOUD and the Photolytic Flow Reactor (PhoFR) results, respectively. Both concluded that the thermodynamics

derived by Hanson et al. [2017] for $H_2SO_4$-$NH_3$ clusters from the bulk flow reactor study of Glasoe et al. [2015] are inaccurate; Hanson et al. [2019] speculated this was due to a cross-contamination by an amine in the ammonia delivery lines. We present here new ternary system measurements from PhoFR and discuss the effects of potential contaminants. We compare predictions of our cluster model to the CLOUD experimental data and to those of Kürten [2019] and Yu et al. [2020]. Finally, we present the effects of relative humidity (RH) on the nucleation rate in the ternary ammonia-sulfuric acid-

water system.

**2. Experiment.**

Nucleation was studied in PhoFR, a vertically-mounted, 120 cm long, 5 cm inner diameter, jacketed, glass flow reactor surrounded by 4 UV (~ 365 nm peak) lamps. The total flow was 2.9 sLpm (273 K and 1 atm, standard L / min), temperature was 296 K, the main gas-source was liquid nitrogen that has ppm-levels of oxygen present, and $H_2SO_4$ was produced from

photolysis of HONO and subsequent oxidation of $SO_2$. Typical flows were: dry nitrogen, $Q_2$, 1.33 sLpm; water was introduced as a humidified $N_2$ flow, $Q_3$, 1.5 sLpm; HONO level was set with a flow, $Q_4$, of between 2 and 10 sccm (standard





$cm^3$ per min); the flow of the $SO_2$ mixture (0.15% in $N_2$ or 1% in 0.1 $O_2$/0.9 $N_2$), $Q_1$, 4 to 32 sccm. For the majority of the work presented here, there was no added base. For some experiments, diluted base in an $N_2$ flow of about 35 sccm was added via a port at the top of the flow reactor, as described previously. The base mixing ratios quoted below were calculated

from known flow rates and assumed fast mixing and no losses; only small portions of the flow reactor achieved those mixing ratios (note that the simulations included mixing and losses). Particles were detected with a nanoparticle mobility particle sizer (MPS) system (DEG system, Jiang et al. [2011]); this and the complete apparatus are fully described in Hanson et al. [2019]. The Supplement details more of the system's essentials.

A second particle detector, the ultrafine condensation particle counter (UCPC, butanol) of Zollner et al. [2012], was attached
to the exit of PhoFR for some experiments. A similar comparison of results from the UCPC and the DEG system found good agreement [Glasoe et al. 2015]. The UCPC is described in Stolzenburg et al. [1991] with the modifications suggested by Kuang et al. [2012]: saturator minus condenser temperature was 35 C and condenser and capillary flow rates were 0.45 L/min and 0.06 L/min, respectively. Pulse-height analysis data was collected from which the total particle number concentrations were obtained. Losses of nanoparticles in sampling lines and inside the UCPC were accounted for using the
Gormley and Kennedy [1948] equations. Activation efficiency is greater than 80 % for 2.5 nm and larger particles [Kuang et al., 2012]. The UCPC sampled flow via one leg of a Swagelok ¼" tee usually with an additional 0.3 L/min transport flow (total sampling rate of 0.75 L/min). The UCPC data was recorded with the tee placed after the DEG system charger; a limited set of early experiments with the tee upstream of the charger did not show large differences in results. Splitting the sample flow with tube fitting tees probably introduces losses but they are on the order of 5%, depending on the details of the
sampling tabulation [Wang et al. 2002]. Since losses are small, activation efficiencies are high, and properly accounting for them requires a large amount of effort, they were not accounted for.

While there may be a ~20% undercount in the UCPC results as detailed in the previous paragraph, this may be counteracted somewhat as the UCPC detects more particles than are in the leading edge of the particle size distributions of the DEG system. It is difficult to quantify this amount because the pulse-height response of the instrument depends on the
composition of the particles [O,Dowd et al. 2004; Hanson et al. 2002].

Growth studies were performed where $H_2SO_4$-dimethylamine nano-particles, formed in the bulk flow reactor of Glasoe et al. [2015], were introduced into PhoFR where they could be exposed to $H_2SO_4$. The growth of the particles was assessed by changes in the particle size distributions upon turning on the UV lights. Specifically, the volume-weighted mean diameters of the leading edge of the size distributions are derived and plotted as a function of HONO. These studies were performed
over a limited range of HONO concentrations because nucleation of new particles at high HONO levels significantly interferes with the initial size distributions.

The source of HONO is the reaction of HCl with NaONO(s) and, as discussed by Febo et al. [1995], NO can be produced by decomposition of 2 HONO molecules into $H_2O$, NO and $NO_2$, possibly followed by reaction of $NO_2$ with HONO giving NO and $HNO_3$. A chemiluminescent NO and $NO_x$ detector (TECO 42b) was periodically used to assess the NO and the
NO+$NO_2$+HONO levels exiting PhoFR. It was calibrated with a 105 ppb NO in $N_2$ mixture (AirGas).



The 2D model of PhoFR developed in Hanson et al. [2019] was also used here. Flow was assumed to be fully developed laminar (parabolic velocity profile) and the full suite of photolytic reactions were included along with $H_2SO_4$ condensation to and evaporation from its molecular clusters. Evaporation rates were determined by thermodynamics for clusters up to ten $H_2SO_4$ molecules; clusters larger than this were not allowed to evaporate. Note that water molecules were not tracked in the clusters but were allowed to affect their size (thus kinetics) and evaporation rates (thus thermodynamics); this quasi-unary approach for simulating the binary system is similar to that of Lovejoy et al. [2004] and Yu [2005].

Note that base (ammonia or dimethylamine) molecules can be included in the simulations in a quasi-binary approach to these ternary systems. Molecular clusters up to ten $H_2SO_4$ molecules containing up to ten base molecules could be simulated; the simulations generally required only up to 3 base molecules (i.e., results were within a few percent of simulations when up to 6 base molecules were included.) The binary (quasi-unary) free energies were allowed to significantly influence the ammoniated cluster free energies for the larger clusters, thus leaving a potential avenue (not yet explored) for an RH dependence for the $NH_3$-$H_2SO_4$ quasi-binary free energies. Further details including kinetic rates, diffusion coefficients, cluster thermodynamics and other assumptions are presented in the Supplement.

## 3. Results and Discussion

### 3.1 System stability, HONO source evaluation

A long time series of data (particle number density, $N_p$, vs. time) is shown in Figure 1 for a set of flow rates that give standard experimental conditions: 52 % RH, 296 K, and a putative [HONO] of $5 \times 10^{11}$ cm$^{-3}$. The previously published $N_p$ data from May and June 2018 [Hanson et al. 2019] are also included. The data after Nov 2018 are generally much lower than the data previous to this which we believe is due to an improvement in the cleanliness of the system.

The data in Figure 1 reveal a long-term stability of the system but with sporadic changes that might be due to contaminants etc.: the spikes and variability in $N_p$ are often associated with cylinder changeovers (+ symbols; those in bold correlate with spikes). Despite repeated flushing of the gas-delivery lines, room air and perhaps dust may get introduced into PhoFR during a changeover; degassing of the exit of the dewar's valve and the regulator port that are exposed to the atmosphere for weeks while at the supplier (Airgas) may play a role. A stainless steel filter was installed on the flow meter manifold in late August 2019 but spikes still appeared after that. Nightmare compounds such as diamines could be responsible as they can lead to a single particle for each molecule [Jen et al., 2016]: a $10^{-15}$ mole fraction could lead to spike of $10^4$ cm$^{-3}$.

Yet for all of 2019 there is a 'floor' of about 1000 particles cm$^{-3}$. The floor of the 2019 data is about 3 % of the average of the particle concentrations in early 2018, a significant drop that we attribute to the depletion or elimination of a source of base molecules within the flow. Extrapolating this floor in the DEG data to lower sulfuric acid, the PhoFR data are now more or less in agreement with the photolytic data from the CLOUD consortium but still somewhat larger than our earlier (Zollner et al. 2012) bulk flow reactor data. We present data below that indicates this floor is heavily influenced by a background level of particles in the DEG system, with the UCPC seeing much lower $N_p$. Nonetheless, this data reveals that



the stability of the system over the time periods of years is suitable for studying nucleation and growth for the binary (sulfuric acid - water) system and that the effects of contaminants can be discerned. See the Supplement for plots of $N_p$ and

$D_p$ vs. $Q_4$ binned in four time periods.

Temporal variability in the floor is still evident which may be influenced by variations in experimental conditions such as the oxidant source. To assess the HONO source, we measured NO and $NO_x$ in the effluent of PhoFR and small levels of NO (and also probably $NO_2$) were present, roughly 10 % of the detected level of $NO_x$ ($NO + NO_2 + HONO$): this data as a function of $Q_4$, the HCl-laden flow over the NaONO(s) powder, are shown in Figure 2.

The total amount of $NO_x$ produced in the HONO source is well-represented by a linear dependence on the flow rate $Q_4$ of the HCl-laden flow. However, NO is not linear with $Q_4$ and it has a higher relative variability than $NO_x$. Fluctuating NO impurity levels will affect the $H_2SO_4$ production rate (see below) and thus $H_2SO_4$ concentrations, contributing to the variability of $N_p$. On average, the fraction of $NO_x$ that is NO ranges from 0.13 at $Q_4 = 2.5$ sccm but decreases to 0.07 at $Q_4 = 10$ sccm. A decreased decomposition of HONO at higher flow rates is expected as the NaONO-vessel (a 5 ml pear-shaped

glass vial) is more rapidly flushed; in fact, Febo et al. [1995] recommend flows of 100's of mL per min to minimize the HONO decomposition rate; they used dry gas to dilute water vapor that also helped slow HONO loss. In our system for studying oxidation of organic compounds, this change was instituted and an additional flow of 50 sccm dry air caused measured NO levels to decrease to a few percent of the $NO_x$ level (Hanson et al. "Growth with SOA", in preparation, 2020). We used our model to assess how HONO decomposition affects $H_2SO_4$ levels and thus $N_p$. Three scenarios were run where

$NO_x$ was attributed to (1) 100% HONO, (2) 80 % HONO / 10 % NO / 10 % $NO_2$, or (3) 85% HONO / 10 % NO / 5 % $HNO_3$. Scenario (3) is a situation where 10% of the HONO decomposes to 5% NO and $NO_2$, followed by $NO_2$ reacting with HONO giving an additional 5% NO and $HNO_3$ ($HNO_3$ is not included in $NO_x$, there could be a 4th scenario, 89.5% HONO and 10.5% NO.) A revised photolysis rate of $6.7 \times 10^{-4}$ $s^{-1}$ was used for HONO (see the next section.)

The centerline $H_2SO_4$ is shown as a function of axial distance down the reactor. Compared to the 100 % HONO simulations,

those with 15-20% HONO decomposition and 10 % NO lead to substantial increases in the rate of oxidation of $SO_2$, boosting $H_2SO_4$ production by up to a factor of two. This has a large effect on the simulated $N_p$, which increases by about two orders of magnitude. The particle sizes would also be affected, where the total exposure of a particle to $H_2SO_4$ along the length of PhoFR can be approximately doubled at 10% NO levels. Further discussion of the modeling and how the calculated $H_2SO_4$ from our previous work compares to the present calculations are contained in the Supplement. Not only

does a small amount of NO accelerate $H_2SO_4$ production but these simulations also reveal that calculated $H_2SO_4$ levels are very sensitive to small fluctuations (6%) in HONO levels when NO is present: average $H_2SO_4$ in scenario (3) is 14 % higher than that in scenario (2).

An effect on $N_p$ due to varying $SO_2$ mixing ratios was found in the measurements presented by Hanson et al. [2019] but this effect is now no longer experimentally present. Standard photochemistry predicted a small dependence, but interestingly the

inclusion of a few ppbv of NO (Supplement and see below) increases the predicted dependence for base-free conditions. The system's increased cleanliness (or perhaps change in the type of contaminant) and the decreased experimental





dependence on $SO_2$ level may be related. The Supplement has more discussion and presents experimental and theoretical $N_p$ as a function of $SO_2$.

**3.2 Growth of nano-particles produced in bulk flow reactor**

A series of experiments were conducted where particles generated in a bulk flow reactor (BFR, [Zollner et al. 2012; Glasoe et al. 2015]) were introduced into PhoFR where they could be grown with photochemically generated $H_2SO_4$. BFR flow was 1.5 sLpm and enough dimethylamine was introduced into BFR's base addition port to induce particle number densities sufficient for the growth measurements (the no-loss amine level was typically 15 pptv). Total flow in PhoFR was maintained at 2.9 sLpm and relative humidity and $SO_2$ concentrations were controlled as before as well as HONO with $Q_4$

varied from 1.5 to 6 sccm. The amount of dimethylamine in the initial particles is not known and it is assumed to have no effect on the photochemical growth of the nanoparticles. A set of size distributions from the DEG system as well as the results of three other growth experiments, one with ammonia swapped in for dimethylamine, are presented in the Supplement.

Two sets of growth experiments are shown in Fig. 4, a plot of the volume-mean diameter of the leading edge particles vs. $Q_4$,

the flow of the HCl-laden flow through the NaONO(s) vessel. The diamonds are with the UV lights on, the circles are lights off (plotted at $Q_4 = 0$; HONO level did not affect the initial size of the particles from BFR.) For one set of conditions the BFR particles were shut off so that the particles nucleated in PhoFR could be measured without interference and their sizes are shown as the orange square and green triangle. The solid red line is representative of the leading-edge diameters from our previous work with PhoFR [Hanson et al. 2019].

The increase in diameter of the external nano-particles has roughly the same dependence on $Q_4$ as do particles nucleated in PhoFR: the data from Hanson et al. [2019] is parallel to the growth data reported here. This indicates that growth conditions are similar in PhoFR whether particles are pre-formed or are nucleated there (at least for those at the leading edge). The good correlations between $Q_4$, the leading edge size of nucleated particles, and calculated $H_2SO_4$ [Hanson et al. 2019] is supported here. Since knowledge of the photo-chemistry in PhoFR has improved dramatically, this correlation is explored

further in the next paragraph.

The increase in the size of the particles is directly linked to their exposure to $H_2SO_4$, and we analyze the data at $Q_4 = 4.2$ sccm in a simplified manner to obtain an average $H_2SO_4$. Fig. 4 shows a change in diameter for the initial 2-to-3 nm particles of 6.5 nm (+/- 1 nm) and the total residence time along the flow reactor axis is 22.7 s (from $Z = 0$ to 125 cm and assuming an axial flow velocity of 5.5 cm/s for fully developed laminar flow.) In the Supplement we adapt the growth equation of

Verheggen and Mozurkewich (2002) by augmenting the first term with a $(1+ d_{SA}/D_p)^2$ factor [Lehtinen and Kulmala, 2003] accounting for the size of the $H_2SO_4$ molecule with diameter $d_{SA}$. This augmentation factor can be viewed as a reconciling of gas-kinetic molecule–cluster collision rates (e.g., McMurry [1980]) with particle–molecule collision rates (e.g. condensational growth rates of Fuchs and Sutugin [1970]). The simplification is to use constant values of the size-dependent terms in equation (S4) (see Supplement for a detailed derivation.) The observed growth at $Q_4 = 4.2$ sccm requires an average


$H_2SO_4$ concentration of $8.6 \times 10^9$ cm$^{-3}$ for the 22.7 s the particles spend in PhoFR. The Supplement includes changes in the size-dependent terms in the growth calculation and the resulting average $H_2SO_4$ is within 5 % of this value.

### 3.3 HONO photolysis rate

This average $H_2SO_4$ concentration along with the measured $NO_x$ and NO was used to estimate the HONO photolysis rate. Firstly, because our earlier UV absorption measurements (supplement of Hanson et al. [2019]) showed the probable presence
of $NO_2$, we assume $NO_2$ produced in the HONO self-reaction did not react further with HONO: the measured $NO_x$ is taken to be 80 % HONO, 10 % NO and 10 % $NO_2$. Using our previous photolysis rate of $8 \times 10^{-4}$ s$^{-1}$, simulations yield an average on-axis $[H_2SO_4]$ of $1.03 \times 10^{10}$ cm$^{-3}$ while the blue profile in Fig. 3, using a photolysis rate of $6.7 \times 10^{-4}$ s$^{-1}$, has an average on-axis $[H_2SO_4]$ of $9.0 \times 10^9$ cm$^{-3}$. Comparing this to the growth-increment-derived average $H_2SO_4$ concentration of $8.6 \times 10^9$ cm$^{-3}$, a HONO photolysis rate of $6.4 \times 10^{-4}$ s$^{-1}$ is inferred, a decrease of 20 % from the photolysis rate of $8 \times 10^{-4}$ s$^{-1}$ derived in our
previous work.

It is not clear that the on-axis $[H_2SO_4]$ is fully representative of the growth conditions for the particles sampled by the DEG system: a more appropriate measure might be the mixing-cup $[H_2SO_4]$ which is a flow-weighted average $[H_2SO_4]$. The mixing-cup average over the 2 L/min sample rate (that covers the middle of the flow reactor from 0 to 1.44 cm radius) is about 10 % less than the on-axis average $[H_2SO_4]$. This would require a 10% larger photolysis rate to get the observed
growth. We split the difference between the two and settled on a photolysis rate of $6.7 \times 10^{-4}$ s$^{-1}$. See the Supplement for plots of simulated data, size and radial distributions of $H_2SO_4$ and particles, mixing-cup concentrations, etc.

We have made two important assumptions regarding $H_2SO_4$ molecules colliding with particles: that the mass accommodation coefficient is unity and there is no van der Waals enhancement of the collision rate. These are reasonable assumptions but are not known with certainty; if not true the estimated photolysis rate could be significantly impacted. We note that some
recent work [Stolzenburg et al. 2020] suggests that the $H_2SO_4$ collision rate with particles is affected by van der Waals forces but we note that the uncertainty is large.

With this new estimate for the HONO photolysis rate, the $H_2SO_4$ concentration at the midpoint of the reactor (Z = 60 cm, R = 0 cm) is $9.3 \times 10^9$ cm$^{-3}$ for $Q_4$ = 4.2 sccm. Note that even though the fraction of $NO_x$ that is NO depends on $Q_4$, the relationship between $Q_4$ and $H_2SO_4$ in PhoFR is linear although not strictly proportional (see the Supplement for model
calculated dependencies.)

### 3.4 UCPC vs. DEG

Shown in Fig. 5 are the results of nucleation studies in PhoFR with total particle number densities, $N_p$, measured with both the DEG system and the UCPC plotted vs. $Q_4$, the flow through the HONO generation system. The two counters are in near agreement for large values of $Q_4$ but below 4 sccm there is a discrepancy that grows as $Q_4$ decreases, reaching over a factor
of ten at $Q_4$ = 2.5 sccm and lower. A potential cause is particles formed in ion-mediated processes, either with ambient ionization in the flow reactor or in the charger for the DEG system: $H_2SO_4$ carried into the charger yield $HSO_4^-$ ions which





can grow to large sizes, perhaps stabilized with water molecules or impurity bases. Note that a very small concentration of ions would lead to a relatively high artifact $N_p$ in the DEG system analysis because large correction factors are applied due to inefficient charging of small neutral particles. Note that the UCPC sampled downstream of the charger and it may have

detected some of these charger-formed ions but they may not contribute significantly to the UCPC's count rate; this may not be the case for particles formed by ambient ionization in the flow reactor as the longer times can lead to detectable $N_p$.

The latter of the two binary ion-mediated nucleation processes considered here, nucleation in PhoFR under ambient ionization conditions, leads to predicted $N_p$ (rate times 5 s; Yu et al. [2020], blue diamonds, and Merikanto et al. [2016], open diamonds) that are highly suggestive that our UCPC data in Fig. 5 was affected by ion processes. The $H_2SO_4$

concentration was taken to be $(Q_4/sccm)*2.28x10^9$ cm$^{-3}$ for the Yu et al. and Merikanto et al. predictions. They compare favorably to the data and it is possible that an ambient ionization rate of 2 ion pairs cm$^{-3}$ s$^{-1}$ (as used here) produced a level of 10 particles per cm$^3$ in PhoFR. The peach line is $N_p$ from simulations using a quasi-unary thermodynamic scheme D52, see below. The blue triangles are the simulated $N_p$ of large negative ions (9 $H_2SO_4$ molecules plus bi-sulfate) using quasi-unary thermodynamics deduced from Froyd et al. (2003) (see Supplement). The Froyd et al. bi-sulfate thermodynamics supports a

role for ion processes in our experiment, although they suggest ion processes are somewhat less active in our system than the theoretical work of Yu et al. and Merikanto et al. suggests.

How does this affect the interpretation of the $N_p$ vs. $Q_4$ results for the nominal binary system? Firstly, the low $Q_4$ points should be disregarded for the binary process, including those from the UCPC as ion-mediated nucleation could have contributed. Also, that the low $Q_4$ data is flawed is apparent upon considering the dependence of nucleation rates on $H_2SO_4$

levels. A power dependency of $N_p$ on $Q_4^6$ is shown in the figure. It is representative of the high $Q_4$ data and comparison of its extrapolation to data at low $Q_4$ highlights that other nucleation processes are in effect at low $Q_4$, even for the UCPC. This is based on the expectation that the power relationship should steepen, at least not be weakened, as $H_2SO_4$ decreases. This is because the critical cluster must increase in size to have a decreased evaporation rate as $H_2SO_4$ lowers. Evaporation rates generally decrease as cluster size increases, as it does in the liquid drop model and in quantum chemistry calculations. Some

experiments also show such behavior (e.g. Ball et al.[1999], Wyslouzil et al.[1991]) as does the simulated $N_p$ labeled D52.

That the low $Q_4$ data for the UCPC also appears to be biased high points to a source of particles other than those due to the binary system, such as the binary ion-mediated process discussed above. Other processes may contribute. For example, dimethylamine at a mole fraction of $1x10^{-16}$ leads to simulated $N_p$ of 800 cm$^{-3}$ at $Q_4$ = 4.2 sccm (see Supplement). Another possibility is that the UCPC detects $H_2SO_4$ and clusters to a small degree: $[H_2SO_4]$ in the $10^9$ cm$^{-3}$ range but detected at an

extremely small efficiency, say $10^{-8}$, could still result in 10s of Hz of count rates (thus 10s of particles / cm$^3$). Other clusters such as the dimer and trimer, although likely present at lower levels, ~$10^5$ cm$^{-3}$, might also give significant counts if efficiencies climb rapidly with size.

Assuming that the UCPC $N_p$ at $Q_4$ = 4.2 sccm in Fig. 5 is indicative of the binary system, an estimate for the binary nucleation rate $J_{bin}$ is 50 cm$^{-3}$ s$^{-1}$, converting $N_p$ to $J_{bin}$ by dividing by an estimated nucleation time of 5 s, the transit time over

a 30 cm length of the reactor centered at 60 cm. This assumes that the majority of the large particles are formed near the





midpoint of the reactor. Particles are also likely to form downstream of this region but they will be somewhat smaller than the leading edge particles. Nevertheless, the sulfuric acid concentration for this $J_{bin}$ we take to be the simulated value at 60 cm and on centerline, $[H_2SO_4] = 9.3 \times 10^9$ cm$^{-3}$. This nucleation rate extracted from the measurements is then $J_{bin} = 50$ cm$^{-3}$ s$^{-1}$ for $9.3 \times 10^9$ cm$^{-3}$ $H_2SO_4$, 52 % RH at 296 K which is lower than some recent binary system (nominal) results: the Zollner et

al. [2012] bulk flow reactor result for these conditions (as extrapolated in Hanson et al. [2019]) is 300 cm$^{-3}$ s$^{-1}$ and the SAWNUC predicted rate is 230 cm$^{-3}$ s$^{-1}$ [Ehrhart et al. 2016], extrapolated from $J_{bin} = 3$ cm$^{-3}$ s$^{-1}$ at $5 \times 10^9$ cm$^{-3}$, 292 K and 38 % RH. See the Supplement for model predicted $N_p$ from SAWNUC and from two 52% binary system thermodynamics used here.

The early bulk studies of Wyslouzil et al. Viisanen et al. and Ball et al. report nucleation rates, extrapolated to the present

conditions, that are lower than the present work; consideration of the uncertainties in the sulfuric acid concentrations mitigates these differences as discussed in Zollner et al. [2012]. Comparisons with the recent photolytic studies by Yu et al. [2017] and Tiszenkel et al. [2019] are difficult as these studies were thought to be affected by amines evidenced by high nucleation rates at very low $[H_2SO_4]$. Note that the 292 K CLOUD binary system measurements reported by Ehrhart et al. [2016] are much higher than the SAWNUC predictions; it was believed to be affected by residual ammonia thus not

considered representative of the putative binary system. We discuss the effect of ammonia on the 292 K CLOUD data below.

The UCPC vs. DEG $N_p$ relationship has not been consistent over time. Data shown in Figs. 5 and 6(b) show that the UCPC $N_p$ is generally lower than the DEG system $N_p$. Yet recent data, e.g. the Fig. 6(a) data for ~ 650 ppt NH$_3$, UCPC $N_p$ are as much as a factor of three greater than the $N_p$ in the leading edge of the DEG system's size distribution; integrating over the

size distribution reduces the discrepancy to about a factor of 2. On another occasion, we noticed that the UCPC $N_p$ decreased considerably during the course of a day's experiment. We think these discrepancies may be related to occasional exposure of the UCPC sampling line to room air that might lead to variable amounts of cluster/particle formation along the sampling line.

**3.5 Addition of base**

Ammonia was added to the system in a port at the top of PhoFR just above the illuminated section, as described in Hanson et al. [2019]. The results from both counters are shown in Fig. 6(a) where $N_p$ is plotted versus the NH$_3$ mixing ratio for $Q_4 = 3.0$ sccm (interpolated data). Upon addition of ammonia to the flow reactor, nucleation increases and $N_p$ climbs at a roughly squared power dependence on ammonia. We performed simulations with NH$_3$-H$_2$SO$_4$ thermodynamics for molecular clusters; two different sets of free energies were used and results are shown as the dashed lines in Fig. 6(a). To illustrate an

effect of a contaminant or other particle formation process not included in our simulations, also plotted are net $N_p$ (x and + symbols) where $N_p$ for zero added NH$_3$ was subtracted from the data.

The model results plotted as the green dashed line use the thermodynamics that fit our 2018 data at 52 % RH (NH3_52, Hanson et al. [2019]); it leads to significant over-prediction probably due to slightly over-strong binding energies. This





scheme was developed from data that was influenced by an arguably higher level of contaminants in the system at that time
(March to June 2018) compared to now. It may have also been affected by not including any HONO decomposition nor
initial NO among other differences in the assumed photochemistry that led to calculated $H_2SO_4$ on average about 18 % lower
than the present calculations yield (see Supplement). The new set of $NH_3$-$H_2SO_4$ cluster thermodynamics for 52 % RH was
developed to better model the current data. The model results using this new set (NH3_D52) are plotted as the dashed
orange line; they secure the lower edge of the range of data and exhibit a similar power dependency, 1.7, on $NH_3$ to that
exhibited in the data.

Comparison to our earlier measurements (Hanson et al. 2019), encompassed by the gray quadrilateral (for $Q_4$ = 2.1 to 4.2
sccm) in Fig. 6(a), reveal that the measurements at the highest ammonia levels are somewhat in agreement while the low
ammonia results are more disparate. The previous measurements have a near-unity power dependency on $NH_3$ which is also
exhibited in many other experiments [Kirkby et al., 2011; Kürten et al., 2016; Benson et al., 2011]. The current
measurements have a power dependency on $NH_3$ near 2 (for net $N_p$) which is also exhibited in theoretical predictions (such
as Kürten et al. [2016].)

Another test of the $NH_3$-$H_2SO_4$ cluster free energies is the variation of $N_p$ with $H_2SO_4$. Shown in Fig. 6(b) are measured $N_p$
from the particle counters vs. $Q_4$ with 360 pptv $NH_3$ added. Note the large decrease in the current $N_p$ from that in June 2018
(230 pptv $NH_3$) [Hanson et al. 2019]. The modeled $N_p$ traverses the current data but shows a larger dependence on $Q_4$ ($Q_4^{5.7}$)
than is exhibited in the data ($Q_4^{3.8}$). This could be due to a deficiency of these thermodynamics and/or the presence of a
contaminant that more heavily influences the measurements at low $N_p$ and $Q_4$.

A synergistic effect between amines and ammonia [Yu et al. 2012; Glasoe et al. 2015] may have affected the measurements
if contaminant amine compounds were present. Although it is difficult to ascertain such an effect in our data it might
provide an explanation for the large scatter in the data in Fig. 6(a) that does not seem to decrease at the highest $NH_3$. This
effect would be of a different nature than the contaminant effect discussed in the previous paragraph. This tightly-wound
story of contaminants is speculative, considering the lack of direct knowledge of the contaminant and the little-understood
synergistic effect that is just recently receiving theoretical scrutiny [Temelso et al.; Wang et al. 2018; Myllys et al. 2019].
The lower envelope of the $N_p$ vs. $NH_3$ data in Fig. 6(a) is adequately represented by the model and the data in Fig. 6(b)
indicate that the effects of other processes probably affected the measurements the most at low $N_p$, using the model results as
a gauge. The high $N_p$ data may very well represent the $NH_3$-$H_2SO_4$ system at 52% RH. This is supported by the Glasoe et
al. finding that the synergistic effect dissipates as the ammonia abundance increases beyond a few hundred pptv $NH_3$.

Clearly, the thermodynamics for the binary system ($H_2O$-$H_2SO_4$) are important for interpreting experimental results. They
provide a reference for assessing whether a contaminant has significantly affected a measured nucleation rate and also for
determining the effect of added base on nucleation rates. Please see the Supplement for more discussion of the binary
system experiments and a comparison of free energies.

While the effects of contaminants in PhoFR have decreased since our earlier measurements were published, the episodic
spikes in $N_p$ (Fig. 1) indicate an intermittent appearance of something that boosts a nucleation process other than the binary





process. Since we cannot ensure this other process has no effect at high $Q_4$, the NH3_D52 thermodynamics must be considered a phenomenological description; at zero ammonia it represents a limit on the pure binary system at 52% RH.

Nonetheless, ammonia's influence on cluster energetics is similar in these two sets of $NH_3$-$H_2SO_4$-52% RH free energies (NH3_52 and NH3_D52). For example, the step-wise free energies for clusters up to 5 acid and 3 ammonia molecules differ by less than 1 kcal/mol. In the supplement is a comparison plot of the step-wise free energy changes for acid addition for both free energies.

We argue that the NH3_D52 free energy landscape can be assumed to be free of the effect of a contaminant and of ion-

mediated processes because these rogue processes operated at low ammonia and $Q_4$. Then NH3_D52 free energies can be used in box model simulations to, in effect, extrapolate our data to the experimental conditions of other data sets. This was done previously [Hanson et al. 2019, Fig. 10] using the NH3_52 free energies to compare to some $NH_3$-$H_2SO_4$ experimental results [Berndt et al. 2010; Dunne et al. 2016]. Part of that plot is reproduced in Fig. 7 focusing on the $[H_2SO_4]=1.5\times10^8$ cm$^{-3}$ data. NH3_D52 free energies result in nucleation rates about an order of magnitude below those calculated with NH3_52.

Our conclusions change little from 2019: the Berndt et al. data (2 to 70 parts per million $NH_3$) and the CLOUD data at > 200 pptv $NH_3$ stand apart from all the previous work in this system and are most consistent with data from PhoFR.

The predicted ternary homogeneous nucleation rates of Yu et al. [2020], J(THN) in Fig. 7, are quite different from the current predictions (which are thought to be free of ionization effects) and both of these differ from the neutral ternary system rates predicted by Kürten [2019] (dashed red line). When ionization rates characteristic of ambient levels due to

galactic cosmic rays (GCR) are present, the Yu et al. [2020] ternary ion-mediated nucleation rates (TIMN) approximate the CLOUD data. Since Kürten et al. [2019] derived neutral cluster free energies to explain the neutral rates measured in CLOUD, they are unsurprisingly close to Yu et al.'s TIMN predictions. Yet the purported chemistries of these two schemes are quite different.

One point to add to the comparisons depicted in Fig. 7 is that the putative binary data from CLOUD, thought to be due to a 4

pptv $NH_3$ impurity, departs significantly (many orders of magnitude) from simulated data using NH3_D52 for 4 pptv $NH_3$. The decreasing discrepancy with NH3_D52 as ammonia increases, gray quadrilateral encompassing the CLOUD data, is reminiscent of the relationship in Fig. 6(a) of our earlier data to the present data. This may indicate a similar situation in the two experimental apparatus where a low-level contaminant (that is now after years of use apparently much lower in PhoFR) that is relatively potent leads to particle formation rates well above what is expected. A synergistic effect may also be in

play here with a strong nucleator such as dimethylamine: our box model calculation using DMA_I thermodynamics [Hanson et al. 2017, 2019] and dimethylamine at $10^{-3}$ pptv yields a nucleation rate of $10^{-3}$ cm$^{-3}$ s$^{-1}$ for the conditions in Fig. 7. A plausible scenario is that the synergistic effect of 4 pptv $NH_3$ boosts this to $10^{-2}$ cm$^{-3}$ s$^{-1}$ and that the synergistic effect increases with $NH_3$ level (or the exceedingly low level of dimethylamine increases slightly as $NH_3$ is increased). By sheer abundance, $NH_3$ could come to dominate the amino-containing moieties in the CLOUD mass spectra (e.g. Schobesberger et

al. 2015; Almeida et al. 2013).



### 3.6 Variation of water content

Relative humidity was varied by adjusting the ratio of humidified to dry $N_2$ flows ($Q_3$, flow through the humidifier, and $Q_2$ dry $N_2$ flow) maintaining a constant total flow rate of 2.9 sLpm. Representative data for a constant HONO level (given by $Q_4$=5.25 sccm) vs. $Q_3$ are plotted in Fig. 8 along with the fit to our previous data at $Q_4$=4.2 sccm [Hanson et al. 2019].

Despite a large decrease in $N_p$ at a given RH between then and now, the variation of $N_p$ vs. humidity has a similar dependence on RH as we observed previously (Hanson et al., 2019): a power dependence of 4.75 is shown in the figure. For two sets of data, the DEG system $N_p$ (leading edge) are also shown with good agreement at high $N_p$ while a bias looks to be present in the DEG results at low RH and $N_p$.

There is a considerable level of variability in the UCPC data that may reflect changes in a contaminant in PhoFR. Yet there

is a sensitivity to the temperature of the room (monitored on the surface of the glass cone at the top of the reactor, $T_{cone}$): on the 22$^{nd}$ of April, the $Q_3$=2.3 sLpm UCPC data had $T_{cone}$ = 23.6 C for the 5x10$^4$ cm$^{-3}$ data point and it was 25 C for the 6x10$^3$ cm$^{-3}$ data. The Supplement has a temperature sensitivity example where time series of $N_p$ and $T_{cone}$ are shown. Another set of RH dependency data for $Q_4$ = 4.2 sccm is also shown in the supplement.

Over a comparable range of RH, Zollner et al. [2012] reported a dependence of $N_p$ on RH to the 5-to-9 power, with the

higher value for RH > 50%. There are significant differences between these two experiments that might affect RH dependencies. For example, Zollner et al. had a bulk source for $H_2SO_4$ and a heated mixing region so that the 6 sLpm total flow and the $H_2SO_4$ it was carrying underwent cooling as particles nucleated. Thus the Zollner et al. $H_2SO_4$ underwent a wide range of hydration due to the 17 K cooling. This change in hydration affects its rate of diffusion and thus loss to the wall; cooling is generally 2 K or less in the present work and there is a steadier hydration level for $H_2SO_4$. Nevertheless, the

scatter in the data shown in Fig. 8 precludes putting much significance on the RH$^{4.75}$ dependence and its comparison to the Zollner et al. RH dependency.

The main conclusion is that $N_p$ is much lower at all RH compared to our previous work [Hanson et al, 2019]. An ion-mediated process may play a small role that affects measurements at low RH and thus $N_p$, but since most of the $N_p$ are much greater than the ~10 cm$^{-3}$ expected for this process, the results are dominated by changes in water vapor. But a contaminant

may have affected the present data as was postulated for the previous data [Hanson et al. 2019] where a level of 0.6 pptv methylamine was consistent with the $N_p$ vs. $Q_4$ data. If methylamine is also present here we estimate that its level has dropped to 0.1 pptv (using the squared power dependence on methylamine found for nucleation rate [Glasoe et al., 2015].) Since the identity and the source of the contaminant are largely unknown we can only speculate here but we point out that a level of dimethylamine of 10$^{-4}$ pptv is shown in the supplement to be consistent with the variation of $N_p$ with $SO_2$. In terms

of RH dependencies, primary alkylamines such as methylamine may be influenced by water content while dimethylamine is not (at least at the 2 pptv level, Hanson et al. [2019].)

We varied RH with added ammonia (120 pptv) to see how water influences nucleation in the $NH_3$–$H_2O$–$H_2SO_4$ system. These data, $N_p$ vs. $Q_3$, are plotted in Fig. 9 (orange symbols) along with the Feb-14 data without ammonia added from Fig. 8.




There is a significant dependence of $N_p$ on RH for the added ammonia case. The enhancing effect of $NH_3$ (ratio of $N_p$ with
added $NH_3$ to the nominal binary case) appears to be less at high RH: a factor of ~ 4 at $Q_3$ = 2 and 2.35 slpm while it is a
factor of ten at $Q_3$ = 0.65 and 1.1 sLpm. This may reflect the overwhelming abundance of $H_2O$ vs. $NH_3$: clusters take what
there is more of by opportunity (entropy) rather than by binding energy (enthalpy). $H_2SO_4$ cluster affinity for $NH_3$ is greater
than it is for $H_2O$, but the stepwise free energy difference is less than a few kcal/mol for clusters containing more than 5
$H_2SO_4$ molecules.

Previous work showed similar behavior. Ball et al. showed a larger enhancement due to added ammonia for measurements
at 5 % RH than at 15 % RH. Benson et al. [2009] also reported ammonia enhancement factors that increased with
decreasing RH but were also dependent on other experimental conditions, nonetheless: for 20 ppbv $NH_3$ and an $H_2SO_4$
concentration of $1.2\times10^9$ $cm^{-3}$, the enhancement factor increased from 20 to 400 when the RH decreased from 8 to 4 %.
Although CLOUD experiments (Kürten et al [2016]; Kirkby et al. [2011]; Almeida et al. [2013]) did not systematically
investigate an RH dependence for the ternary system, Dunne et al. explored the effect of RH on ammonia-sulfuric acid
nucleation in a global climate model and saw an effect of up to 34 % in the number density of 3 nm particles at cloud height.
The Dunne et al. ad hoc RH dependency is shown in Fig. 9 normalized to our measured $N_p$ at $Q_3$=1.05 sLpm (the green
dashed line also brushes our model calculated $N_p$, NH3_D52, at $Q_3$=1.50 sLpm.) It describes the observed increase in
particle numbers with RH fairly well but has difficulties below about 25 % RH.

## 4. Summary and Conclusions

The overarching goal is to improve the accuracy of nucleation rates calculated for atmospheric conditions. Which we
believe we are doing—improving the measurements, identifying specific effects of contaminants, refining the
thermodynamics of clusters—and then testing the results of our approach against previous work through detailed
comparisons.

We have a better understanding of the chemistry within PhoFR. We detected small amounts of NO produced in the source
via the HONO self-reaction which significantly affects the photochemistry. Hard-to-detect amounts (e.g. sub-$10^{-15}$ mole
fraction dimethylamine) of amines affected the measurements but their influence has decreased over time. The deployment
of two different types of particle detectors yielded information on nucleation processes other than the binary $H_2SO_4$-$H_2O$ and
ternary $NH_3$-$H_2SO_4$-$H_2O$ systems. The long-term measurements from PhoFR allowed for discerning the decreasing effects
of a contaminant and perhaps uncovered the influence of ion-mediated nucleation processes [Yu et al., 2020; Lovejoy et al.
2004]. Ion-mediated nucleation may play a role at low nucleation rates in the binary and ternary systems, due to ambient
ionization that affects particle formation in the flow reactor and also potentially within the charger of the nano-MPS (DEG
system).

We presented evidence that the CLOUD data for 292 K and warmer were probably affected by a contaminant or non-binary
system process such as occurs for our low $N_p$ data. We contend that 4 pptv $NH_3$ can't be the complete reason and another





contaminant or other processes than the ternary system may be operating in this CLOUD data set – the model results for this amount of $NH_3$ suggest a much smaller nucleation rate than was reported.

We suggest that $NH_3$ at single digit levels is too little to significantly affect our data even when the apparatus is at its cleanest. Intermittent contaminants at levels that affect the measurements do appear, especially on cylinder changeovers.

We suppose inadvertent dust particles can be introduced into PhoFR or onto surfaces along the gas-supply lines (the liquid nitrogen cylinder port's exposure to room air can also be a factor). Note that when the effect of this contaminant had dissipated, we derived a set of free energies from the data that helped us present limiting nucleation rates for the binary and ternary systems, allowing us to make conclusions regarding contaminants or other nucleation processes.

Our room temperature results provide a more stringent test of the binary system thermodynamics than do the CLOUD

chamber results at low temperatures where the influence of relative humidity and thermodynamics are lessened compared to results at room temperature.

The nucleation rates in the ammonia-sulfuric acid-water system measured in PhoFR have also decreased since our 2018 results, indicating that a decrease in an amine-type of contaminant occurred since then. The free energy scheme for $NH_3$-$H_2SO_4$ clusters developed here has ammonia's influence on the putative binary system about the same as in our previous

work [Hanson et al. 2019] for clusters up to 5 $H_2SO_4$ molecules. The decreasing effect of the contaminant is exhibited in the changes to the putative binary free energies and in the ammonia-containing clusters with 6 or more sulfuric acid molecules. These changes in the largest clusters were sufficient to lead to agreement of simulations with experimental.

The effect of water on nucleation rates was explored and progress was made in decreasing the upper limit nucleation rates for the binary system. The free energies for the $NH_3$-free clusters in the $NH_3$-D52 scheme presented here are within a few

kcal/mole of the modified liquid drop model (SAWNUC) of Lovejoy et al. [2004]. The effect of water when ammonia was present is substantial and indicates that there may be a synergistic effect involving water on the cluster's free energies (and possibly a methylamine contaminant at the 0.1 pptv level.) Interestingly, the Dunne et al. ad hoc equation reasonably explains our measured relative humidity effect at 120 pptv $NH_3$ and $\sim 1 \times 10^{10}$ cm$^{-3}$ $H_2SO_4$. Theoretical work on the effect of water in this ternary system is sparse but initial reports on the effect of water on the small clusters ($\sim 3$ $H_2SO_4$ molecules)

suggests small effects, even a decrease in nucleation as RH increases (see Fig. 9(b) of Henschel et al. [2016]). As we have argued here, it is likely that the effect of water in this ternary system is greatest in larger clusters, those with 6 or more $H_2SO_4$ molecules.

The model's HONO photolysis rate was tuned to give $H_2SO_4$ concentrations that explained the results of the nanoparticle growth studies. Collision rates of SA molecules (whether hydrated or not) with clusters were assumed to be free of van der

Waals interactions and the mass accommodation coefficient $\alpha$ was assumed to be unity [Hanson, 2005]. If van der Waals interactions are important and $\alpha$ is unity then there is a lower level of SA in the flow reactor than the model was tuned to and the thermodynamics are not valid; binding energies are too small. On the other hand, if van der Waals interactions are not important and $\alpha$ is less than unity, then our thermodynamics have binding energies that are too strong. Future work on the growth of larger nano-particles may shed light on this issue. Also, measurements of $H_2SO_4$ uptake efficiency onto particles



smaller than the 140 nm diameter particles that Hanson [2005] studied will provide additional information on the efficacy of van der Waals interactions on collision rates.

The next steps for PhoFR measurements include measuring the particle growth of externally generated nanoparticles as a function of the initial nanoparticle size and composition. These measurements will help tease out information on whether van der Waals interactions are important for uptake of $H_2SO_4$. Coupled with new measurements of the $H_2SO_4$ uptake

coefficient on sulfuric acid particles at sizes smaller than the 140 nm diameter particles used in Hanson [2005], these interactions can be further delineated. We also plan to investigate the effect on nucleation rates of adding methylamine to PhoFR.

We are working on phenomenological free energy schemes for methylamine-$H_2SO_4$ clusters from the BFR results of Glasoe et al. These will be used to compare to measurements in this system and will also be used to estimate a potential level of

methylamine contaminant that could have affected our nominal binary system measurements. We are also working on a phenomenological free energy landscape for $H_2O$-$H_2SO_4$ clusters that can bridge the relative humidity results, effectively providing a basis for our quasi-unary thermodynamics as well as providing composition information.

**Data and Code availability**: Spreadsheets for the data in the figures are available by request. The code for the simulation of
the flow reactor and the box model are written in Delphi and can be made available upon request.

**Author contribution**: SM and DH designed the experiments, carried them out, and analyzed the data. JK and DH developed the model code and performed the simulations. DH, JK and MW evaluated the photo-chemistry and interpreted the nucleation results. DH prepared the manuscript with contributions from JK and MW.

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

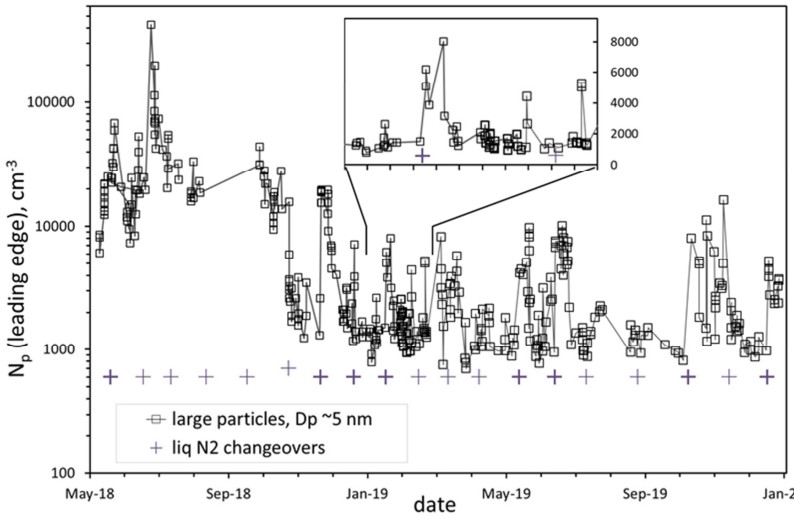

**Figure 1:  Total number density of particles (DEG system) in the leading edge of the distribution over time for standard conditions (Q₄= 4.2 sccm, RH 52%, 296 K).  Leading edge refers to all particles described by a log-normal distribution of the largest particles (roughly 5 nm mode diameter for these conditions).   The approximately monthly liquid nitrogen cylinder changeovers are indicated by the + symbols.  The inset shows a 55 day period of data on a linear scale.**





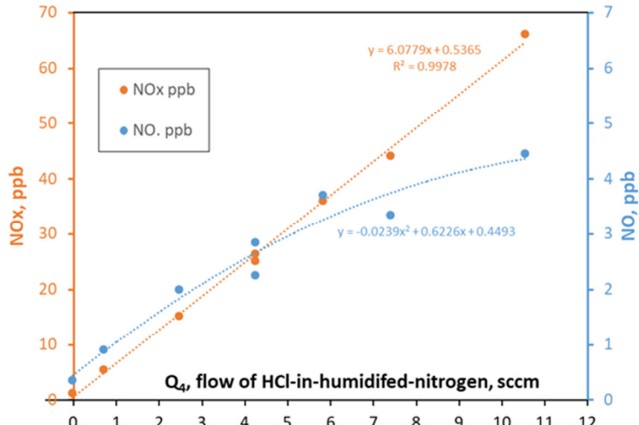

**Fig. 2. NOx (left axis) and NO (right axis) vs. flow rate $Q_4$ of HCl-laden flow over NaONO(s). Assuming that the NOx level is related one-to-one to HCl, the HCl mixing ratio in flow $Q_4$ is 12.8 ppmv. Note: sccm = standard cm³ per min.**


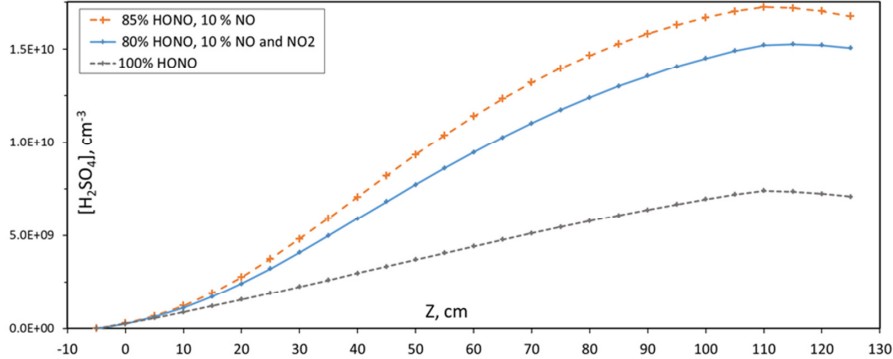

**Fig. 3. Simulated centerline $H_2SO_4$ vs. axial distance for three different HONO decomposition scenarios. Parameters: HONO photolysis rate of $6.7 \times 10^{-4}$ s$^{-1}$, 12.8 ppmv NO$_x$ in $Q_4$ and no reaction between $HO_2$ and $SO_2$. These simulations were run for $Q_4$ = 4.2 sccm. HONO assumed to be 100% of NOx is a level of $4.4 \times 10^{11}$ cm$^{-3}$ or ~18 ppbv.**






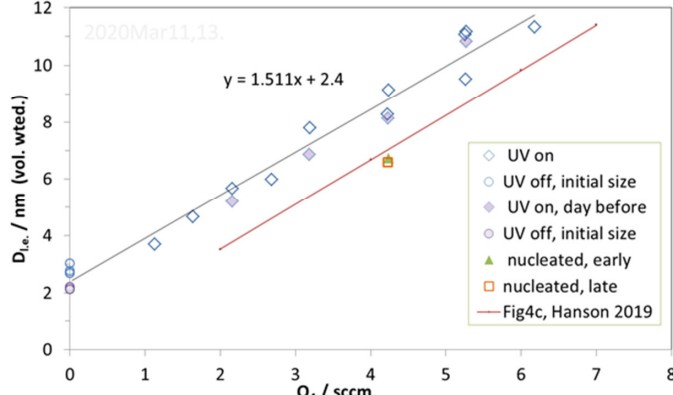

**Fig. 4. Growth studies of externally produced nanoparticles, $D_{le}$ vs. $Q_4$. $D_{le}$ is the volume-weighted diameter of the leading edge of the size distribution. The nanoparticles initial size is indicated by the data plotted at $Q_4 = 0$ sccm. The red line labeled Fig4c is from Hanson et al. [2019].**


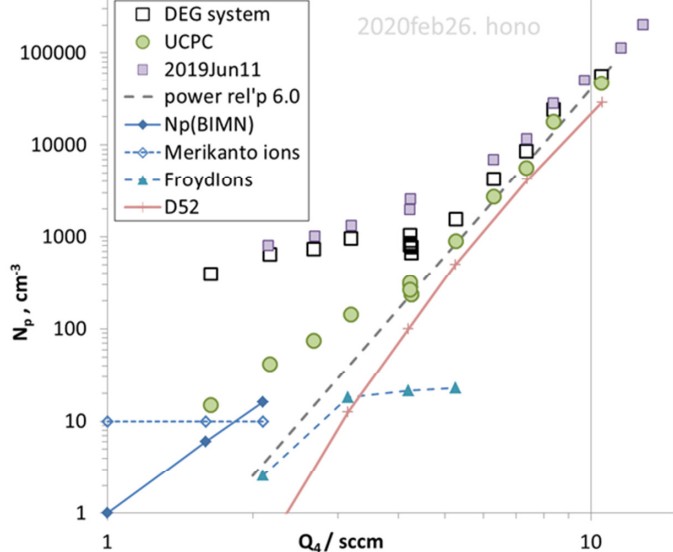

**Fig. 5. $N_p$ vs. $Q_4$, two different particle counting systems. DEG system refers to the nano-MPS followed by a diethyleneglycol CPC and UCPC is a butanol-based detector. DEG system data from 2019 are also shown. Np from nucleation rates $N_p$(BIMN, binary ion-mediated nucleation) from Yu et al. [2020] and from Merikanto et al. [2016] (rates multiplied by 5 s.) $N_p$ calculated in our 2D**
**model for ions (with thermodynamics from Froyd et al., blue diamonds) and for neutral nucleation (peach +).**

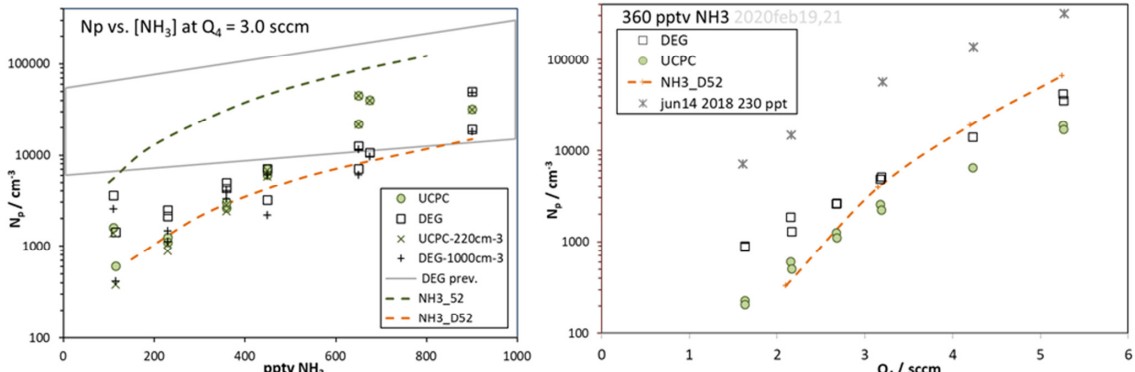

**Fig. 6.** $N_p$ vs. (a) ammonia level and (b) $Q_4$. (a) Experimental data for the two particle detection systems, minus the zero-added base $N_p$, are plotted as the squares (DEG) and diamonds (M1) and the gross $N_p$ are also shown. Model predicted $N_p$ (binary $N_p$ are negligible) are shown as the green (NH3_52) and orange (NH3_D52) dashed lines. Clusters up to 10 acid molecules and either 3 or 6 ammonia molecules were simulated; runs with 6 ammonia molecules were within 3 % of runs with 3. Supplement has further details of the model runs. Gray quadrilateral "DEG prev." encompasses data from Fig. 5 of Hanson et al. [2019]. (b) Symbols as in (a) with $NH_3$ present at 360 pptv. Asterisks are data from 2018Jun14 at 230 pptv $NH_3$.

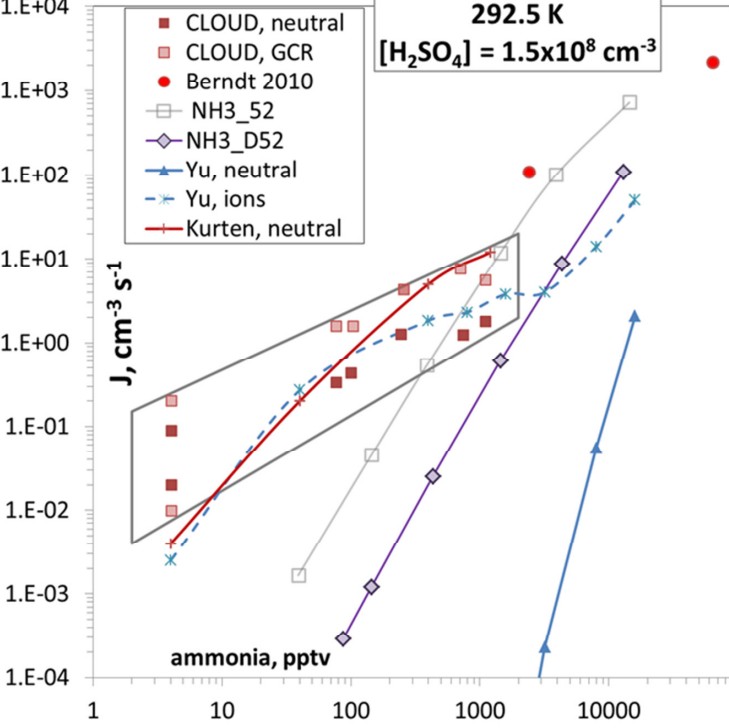


**Fig. 7. Nucleation rate J versus ammonia level for [H₂SO₄]=1.5x10⁸ cm⁻³ and 292.5 K. Berndt et al. 50% RH and CLOUD at 38 % RH (Dunne et al. [2016] also included are low ionization rate measurements, "GCR"; the data at 4 pptv NH₃ were for H₂SO₄ up to 3x10⁸ cm⁻³). Box model calculations (described in Hanson et al. [2019]) using two different thermodynamic schemes for 52 % RH are also shown. Red solid line from Kürten et al. 2019. Ammonia nucleation from Yu et al. [2020] where neutral is the ternary system (THN) and ions is the ternary system (TIMN) with an ion pair production rate of 2 cm⁻³ s⁻¹.**

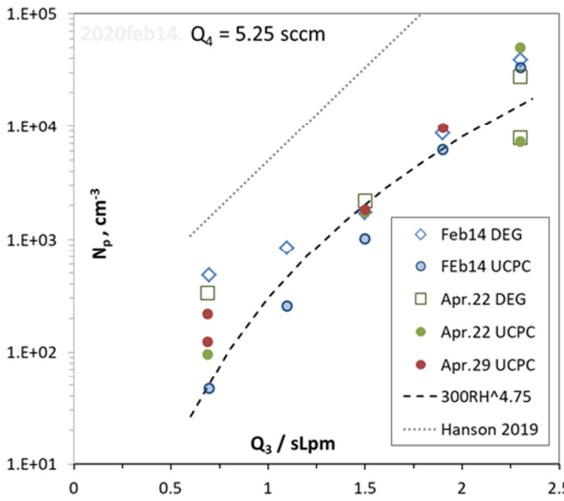

**Fig. 8 Variation of N_p with relative humidity, set by the flow through the humidifier, Q₃. RH ranges from 24 to 81 % for the range of Q₃ in the figure. HONO level was kept constant, Q₄ = 5.25 sccm. A power dependency on RH of 4.75 is shown as the dashed line. The thin dotted line is the exponential fit to the RH-dependency found in Hanson et al. [2019] at Q₄=4.2 sccm.**

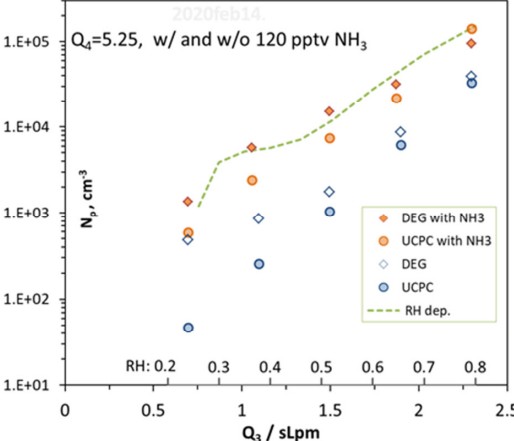

**Fig 9. Variation of N_p with Q₃, a proxy for RH; with and without 120 pptv NH₃ added. Q₄ = 5.25 sccm and N_p from both particle counters are shown. RH varies from 23 to 81 % over the range of Q₃ investigated. The Dunne et al. *ad hoc* RH dependency is shown, normalized to our measurements at Q₃ = 1.05 sLpm.**