# Peer review of "Measurement Report: Sulfuric Acid Nucleation and Experimental Conditions in a Photolytic Flow Reactor"

_Atmospheric Chemistry and Physics, 2020_

## Referee Comment (RC1) · Anonymous Referee #1 · 21 Sep 2020

The measurement report by Hanson et al. (2020) is a follow-up publication on their paper from 2019 (Hanson et al., 2019). In both publications, a photolytic flow reactor (PhoFR) is used to generate $H_2SO_4$ from HONO photolysis and subsequent reactions with $SO_2$, $O_2$ and $H_2O$. For concentrations of $H_2SO_4$ in the $\sim$1e+09 cm-3 range and a reaction time of $\sim$5 s, new particle formation for the binary $H_2SO_4$-$H_2O$ system occurs. The $H_2SO_4$ concentration is calculated from a chemistry model described by Hanson et al. (2019), with a model update in the current study. The newly formed particles are measured with a SMPS (Scanning Mobility Particle Sizer) using a CPC (Condensation Particle Counter) with DEG (diethylene-glycol) as the condensing liquid. This system can measure the particle size distribution starting at diameters of $\sim$1.5 nm

(Jiang et al., 2011). In the current publication a second TSI ultrafine CPC, modified according to Kuang et al. (2012), measures the total particle concentration. From the particle measurements formation rates are derived as a function of the $H_2SO_4$ concentrations. Besides the binary system, further measurements are presented by adding different amounts of $NH_3$. Another set of experiments investigates the dependency of the new particle formation rates on different RH settings. Compared with the previous publication, the current study presents several important new upgrades and results: (1) The cleanliness of the PhoFR has improved. This lowers the baseline particle concentrations for the nominally pure binary system. This is important because contaminants (e.g., $NH_3$ or amines) tend to influence nucleation experiments especially at the warmer (room) temperatures. (2) The effect of NO from the HONO source was included in the chemistry model to calculate $H_2SO_4$, which should lead to more accurate sulfuric acid concentrations. (3) An ultrafine condensation particle counter is used to cross-check the numbers from the DEG-SMPS. The new findings yield a revised set of thermodynamic data for the calculation of new particle formation rates (NPF) in the binary and the ternary system. These chemical systems are globally important for NPF. Overall, I recommend publication of the manuscript by Hanson et al. after they have addressed the comments listed below.

Comments

(1) Section 2: Although the chemistry model is described in the earlier publication by Hanson et al. (2019) it would be good to add a paragraph, which summarizes the chemistry treated by the model.

(2) It is mentioned that the binary nucleation experiments yield the lowest values reported so far. The authors should include a figure, where all their measurements (the earlier ones from 2019 and the current ones) are inter-compared with the results from other studies. Currently such a figure is only shown for the experiments with ammonia but not for the nominally binary system.

[Figure]

(3) In Figure 7 results from a nucleation and growth model are shown for different sets of thermodynamic data. This model is probably rather complex and therefore evaluation would be beneficial. Evaluation could be performed by using an identical set of thermodynamic data and compare the model output to another model. This could, e.g., be done for the ACDC (Atmospheric Cluster Dynamics Code) model together with the thermodynamic data for H2SO4-NH3 nucleation from Ortega et al. (2012). Results for these thermodynamic data using ACDC were presented by Kürten et al. (2016).

Further comments

L155 (page 5): Please specify why NO accelerates the H2SO4 production?

L282 (page 9): Why was the CPC inlet exposed to room air?

References

Hanson, D. R., Abdullahi, H., Menheer, S., Vences, J., Alves, M. R., and Kunz, J.: H2SO4 and particle production in a photolytic flow reactor: chemical modeling, cluster thermodynamics and contamination issues, Atmos. Chem. Phys., 19, 8999–9015, doi:10.5194/acp-19-8999-2019, 2019.

Jiang, J., M. Chen, C. Kuang, M. Attoui, and P. H. McMurry. "Electrical Mobility Spectrometer using a Diethylene Glycol Condensation Particle Counter for Measurement of Aerosol Size Distributions Down to 1 Nm." Aerosol Science and Technology 45 (4): 510–521. doi:10.1080/02786826.2010.547538, 2011.

Kuang, C., M. Chen, P. H. McMurry, and J. Wang. "Modification of laminar flow ultrafine condensation particle counters for the enhanced detection of 1 nm condensation nuclei" Aerosol Sci. Tech. 46: 309–315, doi:10.1080/02786826.2011.626815, 2012.

Kürten, A., Bianchi, F., Almeida, J., Kupiainen-Määttä, O., Dunne, E. M., Duplissy, J., Williamson, C., Barmet, P., Breitenlechner, M., Dommen, J., Donahue, N. M., Flagan, R. C., Franchin, A., Gordon, H., Hakala, J., Hansel, A., Heinritzi, M., Ickes, L., Jokinen, T., Kangasluoma, J., Kim, J., Kirkby, J., Kupc, A., Lehtipalo, K., Leiminger, M.,

Makhmutov, V., Onnela, A., Ortega, I. K., Petäjä, T., Praplan, A. P., Riccobono, F., Rissanen, M. P., Rondo, L., Schnitzhofer, R., Schobesberger, S., Smith, J. N., Steiner, G., Stozhkov, Y., Tomé, A., Tröstl, J., Tsagkogeorgas, G., Wagner, P. E., Wimmer, D., Ye, P, Baltensperger, U., Carslaw, K., Kulmala, M., and Curtius, J.: Experimental particle formation rates spanning tropospheric sulfuric acid and ammonia abundances, ion production rates and temperatures, J. Geophys. Res.-Atmos., 121, 12377–12400, doi:10.1002/2015JD023908, 2016.

McGrath, M. J., Olenius, T., Ortega, I. K., Loukonen, V., Paasonen, P., Kurtén, T., Kulmala, M., and Vehkamäki, H.: Atmospheric Cluster Dynamics Code: a flexible method for solution of the birth-death equations, Atmos. Chem. Phys., 12, 2345–2355, doi: 10.5194/acp-12-2345-2012, 2012.

Ortega, I. K., Kupiainen, O., Kurtén, T., Olenius, T., Wilkman, O., McGrath, M. J., Loukonen, V., and Vehkamäki, H.: From quantum chemical formation free energies to evaporation rates, Atmos. Chem. Phys., 12, 225–235, doi:10.5194/acp-12-225-2012, 2012.
* * *

---

## Short Comment (SC1) · 7 Oct 2020

In this short comment we provide evidence that the referee has reasonably requested. In one of their points they asked for a comparison of our model to the nucleation rates from the ACDC model published in Kuerten et al. [2016].

With the thermodynamics set to those of Ortega et al [2012] (with corrigendum for 4a, 3b cluster) and letting clusters containing 5 sulfuric acid molecules accumulate, we calculated the nucleation rate of 5 acid clusters. This appears to be the nucleated-particles / clusters used for the ACDC model as well (might be 5a and 4b and larger). Please see Hanson et al. [2017] (DOI: 10.1021/acs.jpca.7b00252) section 2.4 and in

its supplement S2.1 and S2.4 for additional information on the 0-D (box) model.

Using the 278 K data presented in Kuerten et al. [2016] for 100 pptv NH3, we simulated three points along the neutral ACDC line (green solid line): SA/cm-3 and J/(cm-3 s-1) of (i) 9e6 and 1, (ii) 1.8e7 and 10, and (iii) 3.8e7 and 100. The results are Jbox/(cm-3 s-1) = 1.2, 12 and 112, respectively. The agreement is good. (*)

We must point out though that using Ortega NH3-H2SO4 thermodynamics limits the simulations to a maximum of 5 acid and 5 base molecules which is problematic for many experimental conditions, particularly for simulations warmer than 278 K. For simulations at 292 K and 100 ppt NH3 and SA at 1e8 cm-3, 5a 5b cluster nucleation is 10.1 cm-3 s-1 which agrees with ACDC as presented in Kuerten et al. If a reasonable extrapolation of the Ortega thermodynamics is applied to the 5a 1-5b clusters such that 6a 1-5b clusters are the accumulation clusters (i.e. the 5a clusters are allowed to evaporate a and b molecules: see plot below) the nucleation rate falls to 1.6 cm-3 s-1. In fact, that is one of the main points of our Measurement Report: that the thermodynamics of the clusters up to ten acid molecules are needed in the NH3-H2SO4-H2O system, even at 278 K - see (*) below.

(*) At 278 K, evaporation rates of the 5a clusters are important. If a reasonable set of free energies are used for the 5a 1-5b clusters, they can evaporate in the model with the accumulation clusters set to the 6a 1-5b clusters. The box model nucleation rates decrease by 32, 64 and 86 % for the (i) through (iii) scenarios, respectively. The trend continues as the thermodynamics are further extrapolated and the accumulation cluster size increases.
* * *
[Figure]

These are step-wise standard Gibbs energy changes for acid (left) and base (right).     Number of base molecules indexed by color.
Any values for 5a or larger or 5b or larger are extrapolated from the Ortega et al. [2012] free energies.

**Fig. 1.** Free energies.

---

## Referee Comment (RC2) · Anonymous Referee #2 · 27 Oct 2020

The article presents upgrades to a photolytic flow reactor system to study nucleation of sulfuric acid and presents some new results using the system. Although sulfuric acid-water (+base) nucleation has been studied extensively by different teams, there are discrepancies in the results obtained using different measurement equipment, so there is a need for validating earlier studies and improving the measurements. Therefore I think the article is in principle worth publication as measurement report. However, the article needs revision for clarity and more discussion on the uncertainties. The authors believe that the differences in their new results compared to their earlier study (Hanson et al. 2019) is due to improved cleanliness of the system, but since there are no actual measurements of the contaminants this remains speculative. Also, the

authors found large discrepancies in the results obtained using two different particle counters. Given the uncertainties, I have doubts how well the results can be compared to other nucleation studies.

General comments:

At first read it was hard to understand the main aim of this measurement report and its connection to the previous study by the same team (Hanson et al. 2019). I think it would be beneficial to state the objectives more clearly in the introduction paragraph.

Chapter 3.4.: I would separate the discussion of why DEG-CPC shows considerably higher counts than UCPC (this should be actually discussed a bit more, see my questions below) from the discussion of which nucleation processes affect the UCPC data (r232-276). To me these seem to be two separate issues each deserving their own chapter.

The summary and conclusions chapter would benefit from shortening and streamlining it. I would concentrate on summarizing what is improved from the 2019 study and what new knowledge that brings, and remove most of the speculation (e.g. related to CLOUD data) that was already discussed in the Results&Discussion part.

As you mention, some recent studies suggest an enhanced collision rate of sulfuric acid molecules (Stolzenburg et al. 2020 but also Halonen et al. 2019) leading to faster growth rates. How much would it affect your results if you include such collision enhancement in your calculations? Can you provide an uncertainty estimation for Fig 7? You note this qualitatively in the conclusions, but maybe this discussion could be moved to results and discussion section and addressed more quantitatively.

Figures: The figure captions and variable names in legends should be revised throughout the article and supplement so that they are self-explanatory. Currently the figures cannot be understood without reading the whole text. E.g. the difference between NH3_52 and NH3_D52 and meaning of M1 (red squares) in Fig S1 are not clear. It

would be helpful if it was made clearer which results are from this study and which are obtained earlier with the same system.

Detailed comments:

I have several specific questions regarding the particle counting, which need to be clarified as it is one of your main new findings, that the DEG system and UCPC show large discrepancies at low $H_2SO_4$ (lower particle concentration).

On p3 r82-85 you write: "While there may be a $\sim$20% undercount in the UCPC results as detailed in the previous paragraph, this may be counteracted somewhat as the UCPC detects more particles than are in the leading edge of the particle size distributions of the DEG system. It is difficult to quantify this amount because the pulse-height response of the instrument depends on the composition of the particles [O,Dowd et al. 2004; Hanson et al. 2002]." The assumed 5% losses for UCPC seem quite low, is there any measurements to characterize the size-dependent losses in the setup used? Does the undercounting depend on the size distribution of the particles you produce, as the two instruments certainly have different detection efficiency curves? Why and how does the composition dependency of pulse-height analysis play a role here, if I understood correctly you use it only to calculate the total concentration?

How often did you measure the background (zero) of your counters, especially the DEG-counter? I'm asking because if you have even very few background counts from homogenous nucleation of DEG, it would be interpreted as large signal in the MPS system. How and how often are these instruments calibrated?

On p9 you speculate about different processes that may affect the concentration measured with UCPC. One possibility brought up is particles formed in a charger (r230) and second the direct detection of sulfuric acid clusters (r254). If your CPC uses pulse-height analysis, shouldn't these particles (which are probably very small at detection) be clearly distinguishable from particles formed in a flow tube?
r277-285 you note that sometimes the relation between UCPC and DEG measured concentration changes (by several factors). To me it sounds the reason has to be technical, as you also speculate. Isn't there any diagnostics you can use to evaluate when one of the counters are measuring incorrectly to eliminate this data? Maybe provide a comparison of the UCPC and DEG measurements in the supplement?

References: Halonen, R., Zapadinsky, E., Kurtén, T., Vehkamäki, H., and Reischl, B.: Rate enhancement in collisions of sulfuric acid molecules due to long-range intermolecular forces, Atmos. Chem. Phys., 19, 13355–13366, https://doi.org/10.5194/acp-19-13355-2019, 2019.

Stolzenburg, D., Simon, M., Ranjithkumar, A., Kürten, A., Lehtipalo, K., Gordon, H., Ehrhart, S., Finkenzeller, H., Pichelstorfer, L., Nieminen, T., He, X.-C., Brilke, S., Xiao, M., Amorim, A., Baalbaki, R., Baccarini, A., Beck, L., Bräkling, S., Caudillo Murillo, L., Chen, D., Chu, B., Dada, L., Dias, A., Dommen, J., Duplissy, J., El Haddad, I., Fischer, L., Gonzalez Carracedo, L., Heinritzi, M., Kim, C., Koenig, T. K., Kong, W., Lamkaddam, H., Lee, C. P., Leiminger, M., Li, Z., Makhmutov, V., Manninen, H. E., Marie, G., Marten, R., Müller, T., Nie, W., Partoll, E., Petäjä, T., Pfeifer, J., Philippov, M., Rissanen, M. P., Rörup, B., Schobesberger, S., Schuchmann, S., Shen, J., Sipilä, M., Steiner, G., Stozhkov, Y., Tauber, C., Tham, Y. J., Tomé, A., Vazquez-Pufleau, M., Wagner, A. C., Wang, M., Wang, Y., Weber, S. K., Wimmer, D., Wlasits, P. J., Wu, Y., Ye, Q., Zauner-Wieczorek, M., Baltensperger, U., Carslaw, K. S., Curtius, J., Donahue, N. M., Flagan, R. C., Hansel, A., Kulmala, M., Lelieveld, J., Volkamer, R., Kirkby, J., and Winkler, P. M.: Enhanced growth rate of atmospheric particles from sulfuric acid, Atmos. Chem. Phys., 20, 7359–7372, https://doi.org/10.5194/acp-20-7359-2020, 2020.

---

## Author Comment (AC1) · 27 Oct 2020

We have discovered an error in the mixing ratio of HCl, and thus HONO, listed in Figure 2's caption; also in the body of the manuscript. The main effect of this error is how it affects the value of the derived first-order photolysis rate coefficient for HONO. This was evaluated by comparing the modeled growth rates based on $H_2SO_4$ concentrations to the measured growth of nanoparticles. Basically, the HONO level increased by ∼35 % and thus the re-evaluated photolysis rate decreased by ∼ 35 %.

The mixing ratio obtained from the slope of the NOx data in Fig. 2 is 18 ppmv. There were a number of measurements proximal in time to those depicted in Fig. 2 and the

average of all these NOx mixing ratios is 17 ppmv. The 12.8 ppmv value was taken from a set of NOx measurements that were performed 9 months earlier (the fraction that was NO was different also). We think that these changes are due to changes in the HCl level from our HCl-source, possibly due to small temperature excursions that may lead to a collection, then a later evaporation, of HCl-H2O droplets on glass surfaces above the main liquid level. Also, the NaONO(s) was exchanged with fresh NaONO(s) powder in Nov. of 2019, that might have helped decrease HONO decomposition to NO etc.

We ran simulations using the higher mixing ratio for NOx (with 80% as HONO, and 10% for each NO and NO2), and found that a photolyis rate of 4.2x10-4 results in $H_2SO_4$ levels within a few percent of those calculated previously and shown in Fig. 3 (blue line). We think this is the appropriate photolysis level to use for data presented in the paper. This does not substantially affect any conclusions in the paper. The data-model comparisons in Figs. 5 and 6 will be run again with the appropriate HCl mixing ratio and proper k_phot and we expect very little change in the quantitative results.

Included is a plot of our periodic NOx and NO measurements for the last 15 months. We plan to show and discuss this figure in the Supplement of a revised version of this paper. We will also model $H_2SO_4$ with conditions for 2019: 13 ppm NOx with 22 % of it present as NO and 22% as NO2 – thus HONO to 56 %.

[Figure]

**Fig. 1.** Plot of NOx, NO, and Dp over time.

---

## Author Response (AR1)

… ….
Overall, I recommend publication of the manuscript by Hanson et al. after they
have addressed the comments listed below.

Comments
(1) Section 2: Although the chemistry model is described in the earlier publication by
Hanson et al. (2019) it would be good to add a paragraph, which summarizes the
chemistry treated by the model.
We will add a few sentences and reference specific sections (2.1 and S7) and tables (S1) where details
are presented in Hanson et al. 2019.

> **Comment [D1]:** Lines 163-166 in the "changes" indicated PDF manuscript.

(2) It is mentioned that the binary nucleation experiments yield the lowest values reported
so far. The authors should include a figure, where all their measurements (the
earlier ones from 2019 and the current ones) are inter-compared with the results from
other studies. Currently such a figure is only shown for the experiments with ammonia
but not for the nominally binary system.
This figure is now included as Fig. 5b (see included figure) that illustrates well how the results from
PhoFR have changed over the course of a few years. A new section '3.5 Nominal Binary Results, Then
and Now' was formed from some of the previous text and a new paragraph describing Fig. 5b.

> **Comment [D2]:** Lines 481-532.

(3) In Figure 7 results from a nucleation and growth model are shown for different
sets of thermodynamic data. This model is probably rather complex and therefore
evaluation would be beneficial. Evaluation could be performed by using an identical set
of thermodynamic data and compare the model output to another model. This could,
e.g., be done for the ACDC (Atmospheric Cluster Dynamics Code) model together with
the thermodynamic data for H2SO4-NH3 nucleation from Ortega et al. (2012). Results
for these thermodynamic data using ACDC were presented by Kürten et al. (2016).
The verification that we presented in our SC comment we will work up as a section in the Supplement.
The reviewer's comment helped us to see that we had not published such a verification step for the box
model; we note that the 2D model had been compared to a commercial CFD result (S2.1.1 of Hanson et
al. [2017]).

> **Comment [D3]:** Supplement section S2.1.

Further comments
L155 (page 5): Please specify why NO accelerates the H2SO4 production?
NO reacts with HO$_2$ that is generated in the OH + SO$_2$ +O$_2$ reaction. Will be parenthetically noted in the
revised text.

> **Comment [D4]:** Line 227.

L282 (page 9): Why was the CPC inlet exposed to room air?
To get a reference pulse height for large particles. Because of shifting baselines due to older electronics,
we decided to not rely on pulse heights for sizing information. Nonetheless, we traced the main reason
for changes in the UCPC vs. DEG system responses to a cylinder changeover.

> **Comment [D5]:** This text has been replaced by lines 463-466 and Supplement sections S1.1.1 and S1.1.2.

> **Comment [D6]:** This new figure has been revised for the new version, including model calculations to give a mid-point nucleation rate (i.e. the reference H2SO4 at 60 cm down the reactor.)

[Figure]

Fig. 5(b) Nucleation rate vs. H$_2$SO$_4$ (SA) concentration. For PhoFR data, nucleation rate given by N$_p$
divided by an estimated 5 s nucleation time. The [SA] concentration was that calculated at 30 cm, both
for the Hanson et al. [2019] data set and for this work (open circles). The filled circles are the nucleation
rates for this work but plotted at the 60 cm calculated [SA] concentration. The red arrows show how
the system has evolved for data taken at Q$_4$=4.25 sccm. Nucleation rates from the CLOUD experiment at
292 and 298 K for nominal binary conditions are also shown.

Additional changes not in direct response to this reviewer's comments (see also the end of response to
referee 2.)

In the Supplement there are a few new figures and a couple figures and paragraphs were removed. The
new figure S9 is a plot of the mode diameter of the nanoDMA system and the NOx measurements over
time. It includes some data from when the DEG system was replaced by a second butanol CPC as the
DEG UCPC became very noisy at times. Since the measurements in the figure are for particles of 5 nm
diameter and larger there are no issues that affect the data presented in this figure.

**NOTE: add 5 to all the line numbers referenced in these reponses.** **Formatted:** Font: Bold

Anonymous Referee #2

The article presents upgrades to a photolytic flow reactor system to study nucleation of sulfuric acid and presents some new results using the system. Although sulfuric acid water (+base) nucleation has been studied extensively by different teams, there are discrepancies in the results obtained using different measurement equipment, so there is a need for validating earlier studies and improving the measurements. Therefore I think the article is in principle worth publication as measurement report. However, the article needs revision for clarity and more discussion on the uncertainties. The authors believe that the differences in their new results compared to their earlier study (Hanson et al. 2019) is due to improved cleanliness of the system, but since there are no actual measurements of the contaminants this remains speculative. Also, the authors found large discrepancies in the results obtained using two different particle counters. Given the uncertainties, I have doubts how well the results can be compared to other nucleation studies.

We hope our detailed responses to the referee's comments below will alleviate many doubts. In this paragraph we provide an alternate perspective on the effects of the uncertainties they have detailed. Our statements about increased cleanliness are definitely speculative but it is the simplest explanation and a common problem in nucleation experiments. The large discrepancies between counters we have argued are two-fold. In one case, the differences have (i) revealed a source of error or artifact in our DEG system that improves the analysis of the present measurements and (ii) this artifact led us to calculate ion-mediated nucleation with our model and these calculations indicate that IMN could play a significant role at low sulfuric acid abundances. The other discrepancy (the UCPC Np being larger than the DEG Np) seems to be completely traceable to a cylinder changeover event where particle counts were elevated for a week or two. We note that another referee asked that comparisons to previous experiments be included in an additional graph which we are glad to do as it highlights how the results from PhoFR have improved. We agree that we should temper our conclusions regarding comparisons with other studies because of the uncertainties in the measurements.

General comments:

At first read it was hard to understand the main aim of this measurement report and its connection to the previous study by the same team (Hanson et al. 2019). I think it would be beneficial to state the objectives more clearly in the introduction paragraph.

Excellent point: the previous abstract lacked focus. We have removed a few sentences and now highlight the changes in the results (about an order-of-magnitude or larger in some cases) that have occurred since our initial publication.

> **Comment [D7]:** e.g. First sentence of the abstract clearly states the point of this report.

Chapter 3.4.: I would separate the discussion of why DEG-CPC shows considerably higher counts than UCPC (this should be actually discussed a bit more, see my questions below) from the discussion of which nucleation processes affect the UCPC data (r232-276). To me these seem to be two separate issues each deserving their own chapter.

This comment prompted a good, hard, long look at the text and led to extensive reworking of the discussion of the UCPC corrections and a new set of figures comparing it to the DEG system is presented in the Supplement.

> **Comment [D8]:** They are now separated into two paragraphs, lines 351-369 (in marked up document).

The summary and conclusions chapter would benefit from shortening and streamlining it. I would concentrate on summarizing what is improved from the 2019 study and what new knowledge that brings, and remove most of the speculation (e.g. related to CLOUD data) that was already discussed in the Results&Discussion part.

We agree and three paragraphs have been removed. We highlight better the changes in the measurements from those presented in our previous publication.

As you mention, some recent studies suggest an enhanced collision rate of sulfuric acid molecules (Stolzenburg et al. 2020 but also Halonen et al. 2019) leading to faster growth rates. How much would it affect your results if you include such collision enhancement in your calculations? Can you provide an uncertainty estimation for Fig 7? You note this qualitatively in the conclusions, but maybe this discussion could be moved to results and discussion section and addressed more quantitatively.

This is an important point that we had quantitatively skirted in the paper. We have prepared a paragraph summarizing the effects on the model predictions and how the cluster energetics need to be modified to achieve agreement between model and measurements.

**Comment [D9]:** A new section, 3.B and table, Table 1, lines 695-714.

Figures: The figure captions and variable names in legends should be revised throughout the article and supplement so that they are self-explanatory. Currently the figures cannot be understood without reading the whole text. E.g. the difference between NH3_52 and NH3_D52 and meaning of M1 (red squares) in Fig S1 are not clear. It would be helpful if it was made clearer which results are from this study and which are obtained earlier with the same system.

We have paid close attention to the captions in the revised version and believe it is clear what has been previously published.

Detailed comments:

I have several specific questions regarding the particle counting, which need to be clarified as it is one of your main new findings, that the DEG system and UCPC show large discrepancies at low H2SO4 (lower particle concentration).

On p3 r82-85 you write: "While there may be a _20% undercount in the UCPC results as detailed in the previous paragraph, this may be counteracted somewhat as the UCPC detects more particles than are in the leading edge of the particle size distributions of the DEG system. It is difficult to quantify this amount because the pulse-height response of the instrument depends on the composition of the particles [O.Dowd et al. 2004; Hanson et al. 2002]." The assumed 5% losses for UCPC seem quite low, is there any measurements to characterize the size-dependent losses in the setup used? Does the undercounting depend on the size distribution of the particles you produce, as the two instruments certainly have different detection efficiency curves? Why and how does the composition dependency of pulse-height analysis play a role here, if I understood correctly you use it only to calculate the total concentration?

Our rambling text here led to a misunderstanding of the corrections and we have extensively revised it to make clear the following information. The 20 % undercount was due to not applying an activation efficiency: we have chosen now to instead actually apply the activation efficiency. The 5 % loss that is not (and won't be) applied is due to the sampling through a sharp right angle. Larger losses exist: we did state that the diffusional loss of nanoparticles was calculated using Gormley-Kennedy and in the revision we list typical losses that range up to 60% at 2.7 nm. According to the literature, the two counters have activation efficiencies that are similar down to about 2.5 nm, which is the lower-limit for the comparisons. The composition dependent activation efficiency is an uncertainty and we suppose they are similar whether DEG or butanol fluids; yet we focus on the leading-edge particles with the DEG system whereas that cannot be readily done with the UCPC data.

**Comment [D10]:** Paragraphs on lines 95-116 have been heavily revised.

How often did you measure the background (zero) of your counters, especially the DEG-counter? I'm asking because if you have even very few background counts from homogenous nucleation of DEG, it would be interpreted as large signal in the MPS system. How and how often are these instruments calibrated?

The DEG system background count rates are measured every day at the start of the measurements and occasionally again at the end. These instruments have not been calibrated but are operated in accord with the literature sources we reference.

**Comment [D11]:** A paragraph in the Supplement has been added, S1.1.

On p9 you speculate about different processes that may affect the concentration measured with UCPC. One possibility brought up is particles formed in a charger (r230) and second the direct detection of sulfuric acid clusters (r254). If your CPC uses pulseheight analysis, shouldn't these particles (which are probably very small at detection) be clearly distinguishable from particles formed in a flow tube?

In principle this is true however we think neither are clearly distinguishable from 'normal' nanoparticles. The charger ions are at much lower abundance than the neutral particles furthermore the pulse-heights for clusters and particles of 3 nm or smaller become very spread out and do not form identifiable peaks in the distributions.

**Comment [D12]:** See lines 110-116

r277-285 you note that sometimes the relation between UCPC and DEG measured concentration changes (by several factors). To me it sounds the reason has to be technical, as you also speculate. Isn't there any diagnostics you can use to evaluate when one of the counters are measuring incorrectly to eliminate this data? Maybe provide a comparison of the UCPC and DEG measurements in the supplement?

We now present a comparison of nanoparticle abundance measurements in the Supplement. The two instruments compare admirably, albeit up to 30% different in one case, over the size range of 3 to 12 nm diameter. Furthermore, we have now traced the anomalously high UCPC measurements to 4 measurement days just after a liquid nitrogen cylinder change. Why this intermittent dust event (or whatever) apparently affected the UCPC instrument more than the DEG system is not known. It may be a clue as to the identity of what caused this (or the other) cylinder-change events.

**Comment [D13]:** S.1.1.1.

**Comment [D14]:** This month long event is now documented in S1.1.2.

Other substantive changes:

See the Comment attached to Fig. 1 but we overlooked the capillary flow rate variation corrections in our original version. It has now been applied and averages about 10%.

The new Fig 5(b) uses $N_p$ from the sum of the particles in the DEG-system distribution as was done for the previous data shown in that figure. The H2SO4 concentration at 60 cm as a function of Q4 was taken from a new figure in the supplement, Fig. S11.

[revised manuscript text omitted]

Subscript

| Page 24: [18] Formatted | Reviewer1 | 12/9/2020 5:22:00 PM |
|---|---|---|

Subscript

| Page 24: [18] Formatted | Reviewer1 | 12/9/2020 5:22:00 PM |
|---|---|---|

Subscript

| Page 24: [18] Formatted | Reviewer1 | 12/9/2020 5:22:00 PM |
|---|---|---|

Subscript

| Page 24: [18] Formatted | Reviewer1 | 12/9/2020 5:22:00 PM |
|---|---|---|

Subscript

| Page 24: [18] Formatted | Reviewer1 | 12/9/2020 5:22:00 PM |
|---|---|---|

Subscript

| Page 24: [18] Formatted | Reviewer1 | 12/9/2020 5:22:00 PM |
|---|---|---|

Subscript

| Page 24: [19] Deleted | Reviewer1 | 12/9/2020 5:22:00 PM |
|---|---|---|

-

| Page 24: [19] Deleted | Reviewer1 | 12/9/2020 5:22:00 PM |
|---|---|---|

-

| Page 24: [20] Deleted | David R. Hanson [2] | 12/4/2020 6:47:00 PM |
|---|---|---|

**Total n**

**Total n**